# Effect of Altitude Gradients on the Spatial Distribution Mechanism of Soil Bacteria in Temperate Deciduous Broad-Leaved Forests

**DOI:** 10.3390/microorganisms12061034

**Published:** 2024-05-21

**Authors:** Wenxin Liu, Shengqian Guo, Huiping Zhang, Yun Chen, Yizhen Shao, Zhiliang Yuan

**Affiliations:** College of Life Science, Henan Agricultural University, Zhengzhou 450046, China; liuwenxin0805@163.com (W.L.); m18739652676@163.com (S.G.); zhp20208090@163.com (H.Z.); cyecology@163.com (Y.C.)

**Keywords:** soil bacteria, forest ecosystem, altitude, community assembly, temperate deciduous broad-leaved forest, ecological niche

## Abstract

Soil bacteria are an important part of the forest ecosystem, and they play a crucial role in driving energy flow and material circulation. Currently, many uncertainties remain about how the composition and distribution patterns of bacterial communities change along altitude gradients, especially in forest ecosystems with strong altitude gradients in climate, vegetation, and soil properties. Based on dynamic site monitoring of the Baiyun Mountain Forest National Park (33°38′–33°42′ N, 111°47′–111°51′ E), this study used Illumina technology to sequence 120 soil samples at the site and explored the spatial distribution mechanisms and ecological processes of soil bacteria under different altitude gradients. Our results showed that the composition of soil bacterial communities varied significantly between different altitude gradients, affecting soil bacterial community building by influencing the balance between deterministic and stochastic processes; in addition, bacterial communities exhibited broader ecological niche widths and a greater degree of stochasticity under low-altitude conditions, implying that, at lower altitudes, community assembly is predominantly influenced by stochastic processes. Light was the dominant environmental factor that influenced variation in the entire bacterial community as well as other taxa across different altitude gradients. Moreover, changes in the altitude gradient could cause significant differences in the diversity and community composition of bacterial taxa. Our study revealed significant differences in bacterial community composition in the soil under different altitude gradients. The bacterial communities at low elevation gradients were mainly controlled by stochasticity processes, and bacterial community assembly was strongly influenced by deterministic processes at middle altitudes. Furthermore, light was an important environmental factor that affects differences. This study revealed that the change of altitude gradient had an important effect on the development of the soil bacterial community and provided a theoretical basis for the sustainable development and management of soil bacteria.

## 1. Introduction

Bacteria are ubiquitous soil microorganisms with a wide global distribution and with the largest number and groups of organisms [1]. They serve as a driver in the nutrient cycling of forest ecosystems and are also a good indicator of spatial pattern changes [2]. Thus, understanding the distribution patterns of soil bacterial communities along altitude gradients and the factors driving these patterns has important implications for maintaining ecosystem processes and functions [3]. Currently, many uncertainties remain about how the composition and distribution patterns of bacterial communities change along altitude gradients, especially in forest ecosystems with strong altitude gradients in climate, vegetation, and soil properties. Therefore, exploring the variation of bacterial diversity along the altitude gradient is useful for comprehensively understanding the altitude pattern of microbial diversity.

The diversity of soil bacterial communities is driven by a combination of multiple factors, and their dynamic characteristics and sensitivity to environmental changes may affect their ecological functions [4,5]. Altitude gradients are considered one of the key factors influencing the structure and diversity of soil bacterial communities. In mountain ecosystems, these gradients are characterized by variations in environmental factors, such as temperature, humidity, light, vegetation types, and soil properties, which collectively affect the composition and diversity of soil bacterial communities [6,7]. Many studies have documented the changing patterns of soil bacterial diversity and community assembly along an altitude gradient. For example, Wu et al. reported a decreasing diversity in soil bacterial communities in subtropical forests [8]. Shen et al. observed that the bacterial diversity in Changbai Mountain showed no significant trend with increasing altitude [9]. Dharmesh et al. found that the rise of soil bacterial diversity along an altitude gradient increased first and then decreased [10]. These differences may be related to differences in physicochemical soil properties, climate, and light between the study areas. To further understand the altitude pattern changes of soil bacteria, studies are needed on the variation patterns of soil bacterial diversity along an altitude gradient in forest ecosystems. Currently, which environmental factors shape the altitude landscape of bacterial communities in mountain ecosystems remains poorly understood.

Vegetation types change along altitude gradients, thus affecting soil properties and temperature and bacterial community development and diversity [3]. Some studies have pointed out that soil organic matter content also changes with altitude, thus affecting soil suction community assembly and function [8,11]. Temperature is the main environmental factor that influences elevational patterns of bacterial diversity in the Andes, whereas soil pH is a key driver in the distribution of forest soil bacterial communities along an altitude gradient [10,11,12]. These studies suggest that the community composition and distribution pattern of soil bacteria along an altitude gradient may be strongly influenced by environmental factors such as soil physiochemical properties and temperature. Further research is needed to study the distribution pattern of soil bacteria along altitude gradients as well as the mechanism of environmental factors. Therefore, exploring the environmental factors driving the altitude patterns of soil bacterial communities is important for an improved understanding of the altitude patterns of soil microorganisms.

Understanding the ecological processes of soil bacterial communities is a major challenge [13]. In general, ecological processes are divided into deterministic and stochastic processes. A stochastic process refers to unpredictable events in an ecosystem governed by random factors, whereas a deterministic process is characterized by predictable and systematic influences from ecological factors. Most studies have shown that bacterial communities are influenced by deterministic (niche theory) and stochastic processes (neutral theory) [14]. In addition, studies have shown that environmental factors can regulate the balance between deterministic and stochastic processes [15]. For example, soil pH and water content are the main drivers of the balance between deterministic and stochastic processes regulating rich and rare bacterial subcommunities, respectively [15]. Low salinity, for example, contributes to the predominance of stochastic processes in microeukaryote plankton [16]. However, the key environmental factors that regulate the balance of community-building mechanisms between speciated and generalized species in soil bacterial habitats remain an open question.

The change of topography and climate on Baiyun Mountain leads to the obvious vertical division of forest types on Baiyun Mountain. This paper aims to explore the distribution rules of soil bacterial subcommunities (specialized species, neutral species, and generalized species); analyze their diversity and community structure characteristics; understand the ecological processes of these bacterial communities on different altitude gradients; and explore the changes of soil bacterial communities and subcommunities on these elevation gradients.

## 2. Materials and Methods

### 2.1. Study Site

The Baiyun Mountain Forest National Park of Henan Province (33°38′–33°42′ N, 111°47′–111°51′ E) is located in the hinterland of the Funiu Mountain in the south of Song County, Luoyang City, Henan Province, China (Figure 1). The mountain is high in the west and low in the east, and it extends gently from north to south, with a slope of mostly 40–80° [17]. The Park belongs to the subtropical warm temperate climate transition zone, and the average altitude of this area is 1500 m, with a forest coverage rate of more than 95%. Furthermore, it consists of 1991 species of plants, including *Quercus aliena* var. *acutiserrata*, *Toxicodendron vernicifluum*, *Quercus variabilis*, *Larix gmelinii*, *Sorbus alnifolia*, *Corylus heterophylla* [18]. The average annual rainfall is 1200 mm, mostly from July to September. The annual average temperature is 18 °C, and the highest temperature does not exceed 26 °C. Brown and dark soil are the main soil types, with pH a value of 5.5–6.5, which is acidic.

### 2.2. Environmental Data Collection

In accordance with the construction standard of the CTFS, a long-term fixed monitoring point of 240 m and 200 m of 4.8 hm^2^ was established in the Baiyunshan Forest National Park [19]. The long-term fixed monitoring plots of 4.8 hm^2^ were divided into 120 20 m × 20 m samples (400 m^2^ per sample square) (Figure 1). Three soil samples were collected from each 20 m × 20 m subsample (with a distance of 10 m between each two sampling sites). During sampling, impurities such as stones, litter, and roots, were removed, and then the soil layer with a surface up to 10 cm deep was collected. Afterwards, the three soil samples were evenly mixed as the mixed soil sample of the small sample. A total of 120 soil samples were collected. Part of the collected soil was loaded into a sterile self-sealing bag, stored in an ice box, and brought back to the laboratory in a −80 °C refrigerator for Illumina sequencing. The other piece of soil was air-dried and ground through a 2 mm sieve to remove debris for the determination of soil physical and chemical properties.

Hemispheric photographs were taken with a Canon SLR camera (EOS60D, Canon, Tokyo, Japan) with an ultrawide lens (SIGMA, St. Louis, MO, USA, 4.5 mm F 2.8 EXDC) at four corners of the 20 m × 20 m square samples 1.3 m above the ground. To avoid inaccurate data, photos were taken at dawn, dusk, and on cloudy days [20]. To ensure a similar contrast during the day, different exposure times (1/60 s, 1/125 s, 1/250 s) were used, and then the photo with the highest contrast between the sky and leaves was selected as the valid photo. The selected valid photographs were processed by the woodland canopy digital analysis system. Average leaf angle (ALA), canopy coverage (CC), total radiation (TR), scattered radiation (SR), direct radiation (DR), light transmittance (LT), and leaf area index (LAI) were obtained.

The calculation of topographic factors is a common method of the ForestGEO (Forest Global Earth Observatory) plot [21]. Mean altitude (MEA), convexity (CON), slope, (SLO), and aspect (ASP) were measured per 20 m as described by Harms et al. (2001) and Valencia et al. (2004) [21,22].

### 2.3. DNA Extraction and PCR Amplification

The total soil bacterial DNA was extracted from 0.5 g of fresh soil samples using the FastDNA SPIN kit (Mobio Laboratories, Carlsbad, CA, USA) according to the manufacturer’s instructions for use [1]. The purity and concentration of the extracted DNA were checked with a spectrophotometer (Nanodrop Technologies, Wilmington, DE, USA), and the quality and integrity of the DNA were determined by 1% agarose gel electrophoresis. The purified DNA was used as a template to amplify the target fragment using standard primers 515F (5′-GTGCCAGCMGCCGCGG-3′) and 806R (5′-GGACTACVGGGTWTCTAAT-3′) for the 16S rDNA V3–V4 region [23]. The amplification period of PCR was 95 °C; primary degeneration was conducted for 3 min in 30 cycles at 95 °C, 30 s; 55 °C, 30 s; 72 °C, 45 s; and finally, 10 min amplification was performed at 72 °C [24]. Amplification products were separated by 2% agarose gel electrophoresis, and PCR products were purified and quantified using a DNA gel extraction kit. All libraries were sequenced for the purified amplicons using the paired-end (2 × 150 bp) method on the Illumina HiSeq platform (Illumina Inc., San Diego, CA, USA) [25]. Sequencing and bioinformatic services were completed by Shenzhen BGI Technology Co., Ltd. (Shenzhen, China).

### 2.4. Bioinformatics Analysis

The initial step involved assembling the raw paired-end FASTQ sequences using FLASH (v1.2.11) with default settings [26]. Subsequently, the resulting raw sequence data underwent comprehensive analysis and processing within the Quantitative Insights into Microbial Ecology pipeline [27]. A total of 7,329,751 sequences were obtained from the bacterial samples. The bioinformatics processing of these reads entailed merging forward and reverse paired reads to generate robust amplicons, utilizing Vsearch with a minimum overlap of 100 nucleotides and a merge read size ranging from 70 to 400 nucleotides [28]. Subsequently, OTU clustering was carried out at 97% sequence identity, followed by rigorous quality filtering through chimeric read removal using the UCHIME algorithm; the UCLUST algorithm was used to classify the sequences into distinct operational classification units (OTUs) [29,30]. Species annotation was conducted using the Greengenes database, which is accessible at http://greengenes.lbl.gov/ (accessed on 13 May 2023).

### 2.5. Analysis of the Habitat’s Generalists, Specialists, and Neutral Taxa

Niche breadth was assessed following the method outlined by Pandit et al. [31], utilizing the Levins niche breadth index [32]:$$B_{j}=\frac{1}{\sum_{i=1}^{N}P_{ij}^{2}}$$

The niche width of each OTU (denoted as *B_j_*) within the communities was calculated, with *P_ij_* representing the relative abundance of OTU j in a specific habitat (in this case, each of the 120 samples was treated as a separate “habitat”) [31,33]. A higher B value for a particular OTU signifies a broader niche breadth. OTUs with broader niche breadths are characterized as being more evenly distributed and metabolically adaptable compared which those with narrower niche widths [34]. The “Niche.Width” function from the R package “Spaa” was used for this analysis [35].

Microbial communities were categorized into generalists, specialists, or neutral taxa based on the Levins niche width [36]. The occurrence of OTUs was determined through the generation of 1000 permutations using the quasiswap permutation algorithms provided by the EcolUtils R package (4.0.1) [37]. OTUs were subsequently classified as generalists, neutral taxa, or specialists based on their occurrence, utilizing permutation algorithms as implemented in EcolUtils. Generalists were identified by having wider fundamental niches compared with specialists [38]. In this study, an OTU was considered a generalist or specialist depending on whether its observed occurrence exceeded the upper 95% confidence interval or fell below the lower 95% confidence interval. OTUs were designated as neutral taxa if their observed niche breadth fell within the 95% confidence interval range [39].

### 2.6. Data Analysis

The Shannon–Wiener index for all samples were calculated using the diversity function in the “Vegan” package (4.0.1) [37]. The Kruskal–Wallis test was used to compare the bacterial diversity index and niche width index in different habitats to explore whether significant differences existed between different groups. Multiple regression tree analysis was used to divide the altitude gradient, and this analysis classified the samples based on the “mvpart” program package of R 4.0.1 [40]. Redundancy analysis was used to explore the effects of environmental factors, such as woody plants, light, topography, and soil, on bacterial community assembly. The analysis was conducted using the Vegan program package in R 4.0.1; whether the effect of each environmental factor on the bacterial community distribution reached significance using the “envfit” function was also tested with 999 [41]. We performed centered log-ratio transformations on the obtained sequencing data and evaluated it as compositional data. Then, t-distributed stochastic neighbor embedding (T-sne) was used to analyze the changes in bacterial community structure in four categories under different altitude gradients [42].To predict the potential importance of stochastic processes in the composition of soil bacterial communities, a neutral community model (NCM) was used, and the method of nonlinear least squares was applied to generate the best fit between the frequency of occurrence and relative abundance of OTUs. The R-value indicates the goodness of fit to the model, which was calculated following “Ostman’s method” [43,44]. Model computations were performed using R version 3.6.1. The effect of deterministic or stochastic processes on bacterial community composition was tested by examining the deviation of each observed indicator from the mean of the null model [45]. The obtained values were standardized to allow comparison between combinations using the standardized effect size.

## 3. Results

### 3.1. Division of the Different Elevational Gradients

The altitude gradients were divided using a multiple regression tree. The division results showed 49 sample squares for the low altitude gradient, 27 for the middle altitude gradient, and 44 for the high altitude gradient (Figure 2).

### 3.2. Species Composition of Soil Bacteria at Different Altitude Gradients

In accordance with the niche width index, 767 OTUs were defined as generalized species, 6753 OTUs were defined as neutral groups, and 5332 OTUs were defined as specialized species. Statistical analysis of the bacterial community assembly of the four taxa by T-sne (Figure 3A) showed that the species composition of the four taxa differed in the three altitude gradients. In addition, most OTUs of all species (77.82%), neutral species (77.06%), and specialized species (75.58%) were shared across the three altitude gradients (Figure 3B), and the proportion of OTUs shared by all groups and neutral groups was higher among all taxa. The results of the Kruskal–Wallis test showed (Figure 3C) that the bacterial richness of all species and specialized species was significantly different (*p* < 0.05), and the bacterial richness was highest at a low altitude and low at high and medium altitudes, where the bacterial richness of neutral groups and generalized species was not significant (*p* > 0.05).

### 3.3. Ecological Processes of Bacterial Communities across Different Altitude Gradients

An NCM successfully estimated most of the relationships between the frequency of OTU and its relative abundance changes, which showed 83.9%, 82.4%, and 83.8% of bacterial community variation in low, medium, and high altitudes, respectively. This result indicates that bacterial communities at different elevation gradients are mainly controlled by stochastic processes (Figure 4A). Furthermore, the niche breadth of bacterial communities at low and high altitudes was significantly greater than that at middle altitudes (Figure 4B). According to the C-score (Figure 4C), the standardized effect size (SES) increased and then decreased along the altitude gradient, indicating that the relative contribution of deterministic processes to the composition of soil bacterial community increased and then decreased with the change of the altitude gradient.

### 3.4. Effects of Environmental Factors on the Soil Bacterial Communities

The community composition of specialized species, neutral species, and generalized species were significantly affected by altitude, slope, aspect, convexity, leaf area index, light transmittance, available phosphorus, soil pH, alkaline nitrogen, soil pH, and water content (*p* < 0.05, Figure 5). The VPA results (Figure 6) showed that 56.43%, 54.35%, and 39.91% of the changes in the entire bacterial community in the high-, medium-, and low-altitude gradients were explained by topography, light, and soil. In the high-altitude gradient, light, soil, and topography explained 17.61%, 14.3%, and 11.76% of the community variation, respectively; in the medium-altitude gradient, light, soil, and topography explained 32.63%, 14.18%, and 8.62% of the community variation, respectively; in the low-altitude gradient, light, soil, and topography explained 12.79%, 10.69%, and 6.64% of the community variation, respectively. Thus, light availability can explain more community variation of bacterial communities than soil and terrain.

## 4. Discussion

Altitude gradient diversity is one of the most basic patterns of soil bacterial biogeography. Bacterial community composition varies with altitude, suggesting that certain factors may play an important role in the altitude distribution of bacterial communities [46]. This study showed a monotonically decreasing increase in bacterial diversity along the altitude gradient in forest ecosystems, which is consistent with the study of Luo et al. (2020) and Bahram et al. (2012) [47,48]. As the most important environmental factor, altitude can affect bacterial community diversity by regulating microclimate, temperature, water, and soil nutrients [49]. The decrease in bacterial richness along the altitude gradient may be due to the influence of soil carbon and nitrogen content [50]. Some areas have unique microbial ecosystems due to extreme environmental conditions, such as high altitude, low oxygen pressure, high solar radiation, and high salinity. The environmental high-altitude wetland conditions were considered drivers of diversification, generating ecological niches for microbialites, which represent a higher degree of microbial complexity [51]. The study found that eukaryotic richness was highest at the most extreme sites, whereas combined bacterial and archaeal richness was highest at less extreme sites [52]. Soil bacteria are more sensitive to soil environmental changes, especially given that soil pH is also a major factor affecting bacterial diversity [53]. Soil bacterial richness for all species, generalized species, specialized species, and neutral groups varied across different elevational gradients, and this difference is largely influenced by woody plants and soil physicochemical properties. Significant differences are found in the physicochemical properties of woody plants and soil under different altitude gradients [31,54]. The lower diversity of specialized bacterial species at high altitudes may be due to harsh environmental conditions (including higher solar radiation, large daily temperature fluctuations, high wind exposure, etc.) [14]. The results of this study showed that the diversity of specialized bacterial species was not significantly varied under different altitude gradients probably because specialized species had higher fitness under certain environmental conditions, whereas generalized species had consistent fitness across gradients of different conditions [55,56,57]. Therefore, the difference in the altitude distribution of specialized species in different habitats was greater than that of generalized species.

The results of this study showed that the elevation gradient is important for soil bacterial community composition, mainly affecting the balance between deterministic and stochastic processes. Bacterial communities exhibit wider niche breadth and stochasticity processes at low altitudes, meaning that community composition is most influenced by stochasticity because deterministic processes probably tend to have less effects on species with wider niche breadth than species with narrower niche breadth. The NCM showed that the migration rate (m = 0.4894) was lower than the high (m = 0.5584) and middle altitudes (m = 0.569). Therefore, the diffusion capacity of the bacterial flora at low altitudes is weak and diffusion-limited, making the bacterial community more susceptible to stochastic processes (diffusion limitation) at low altitudes [58]. Moreover, the C-score results demonstrated the importance of stochastic changes in maintaining ecosystem stability [59,60], and the SES values are larger at middle altitudes, indicating that bacterial community assembly is more influenced by deterministic processes at middle altitudes. The results of this study suggest that stochastic processes are more prominent at low altitudes probably because the high dispersal potential of bacterial communities is associated with habitat homogeneity under similar conditions [61]. In addition, neutral theory suggests that higher abundance and diversity of species has a greater dispersal potential, which is consistent with the conclusion that the bacterial Shannon index is higher than the high/middle altitudes in the present study [62].

Altitude, slope, convexity, organic matter, and alkaline nitrogen were the main environmental factors affecting the whole bacterial community, whereas altitude, slope, aspect, convexity, leaf area index, transmittance, quick phosphorus, organic matter, alkaline nitrogen, soil pH, and water content were significantly associated with the community composition of specialized species, neutral groups, and generalized species. These drastic changes in environmental variables may all lead to changes in the species number and composition of all species, specialized species, neutral groups, and generalized species. As a significant influencing factor common to all three groups, soil pH is one of the key factors affecting the bacterial community structure, which is consistent with other studies [9,63]. Organic matter, water content, altitude, alkaline nitrogen, and quick phosphorus were also found to be significant influencing factors that affect the bacterial community, which is consistent with other studies [64,65]. Moreover, light factor, leaf area index, and light transmittance can significantly change the bacterial diversity and community assembly composition [66]. Differences in light, soil, and topography that significantly affect bacterial specialized species, generalized species, and neutral species may reflect differences in their ability to adapt to their environment. VPA showed that light effects explained the maximum variation in bacterial community composition across altitudes, whereas direct effects of soil and topography had some effect on bacterial community variation. Moreover, forest canopy coverage is the main factor affecting soil bacteria distribution [67]. Studies have found a weak indirect relationship between forest canopy structure and bacterial richness, and different forest canopy results lead to different light availability under the forest, which then affects the bacterial diversity [68,69]. By contrast, the difference between altitude and light can have a large effect on air moisture and soil moisture in the plot, leading to differences in species abundance of different bacterial taxa. In this study, canopy structure had a large effect on bacterial taxa at different altitudes; thus, the light, soil and topography driving the bacterial distribution pattern differed across altitude gradients.

## 5. Conclusions

This study provides an important basis for explaining the bacterial community pattern in the Baiyunshan forest ecosystem based on niche width characteristics. The results reveal significant differences in soil bacterial community composition across different altitude gradients. At different elevation gradients, stochastic processes play a more important role in the overall bacterial community composition process in comparison with deterministic processes. Major environmental factors affecting the entire bacterial community and other taxa vary in different altitude gradients. Altitude, slope, convexity, organic matter, and alkaline nitrogen emerged as significant influencing factors on the overall bacterial community (*p* < 0.05). Meanwhile, the composition of specialized species, neutral groups, and generalized species were notably impacted by altitude, slope, convexity, organic matter, and alkaline nitrogen, as well as by variables such as leaf area index. Light is the main environmental factor driving the variation of the whole bacterial community and other taxa across different altitude gradients.

## Figures and Tables

**Figure 1 microorganisms-12-01034-f001:**
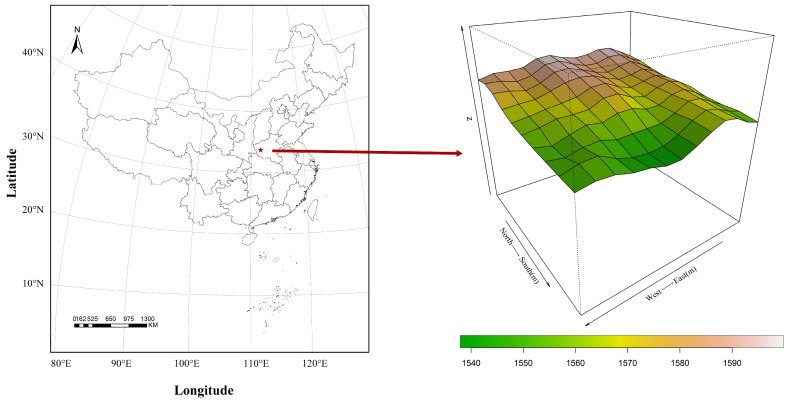
Topographic map of the 4.8 hm^2^ fixed monitoring plot in the Baiyun Mountain Forest National Park.

**Figure 2 microorganisms-12-01034-f002:**
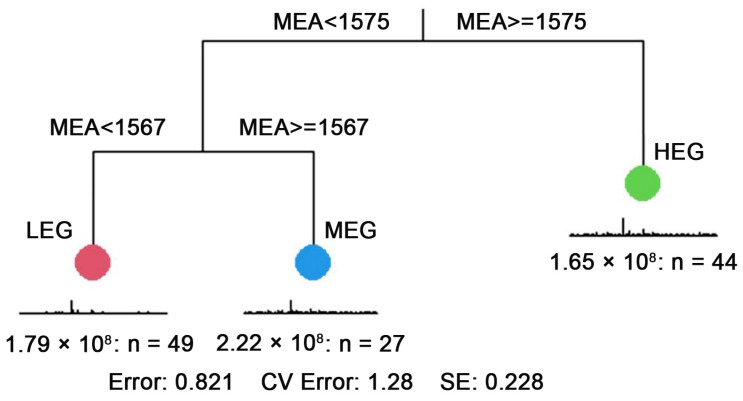
Multiple regression tree analysis of altitude variables. Note: HEG: high altitude gradient; MEG: medium altitude gradient; LEG: Low altitude gradient.

**Figure 3 microorganisms-12-01034-f003:**
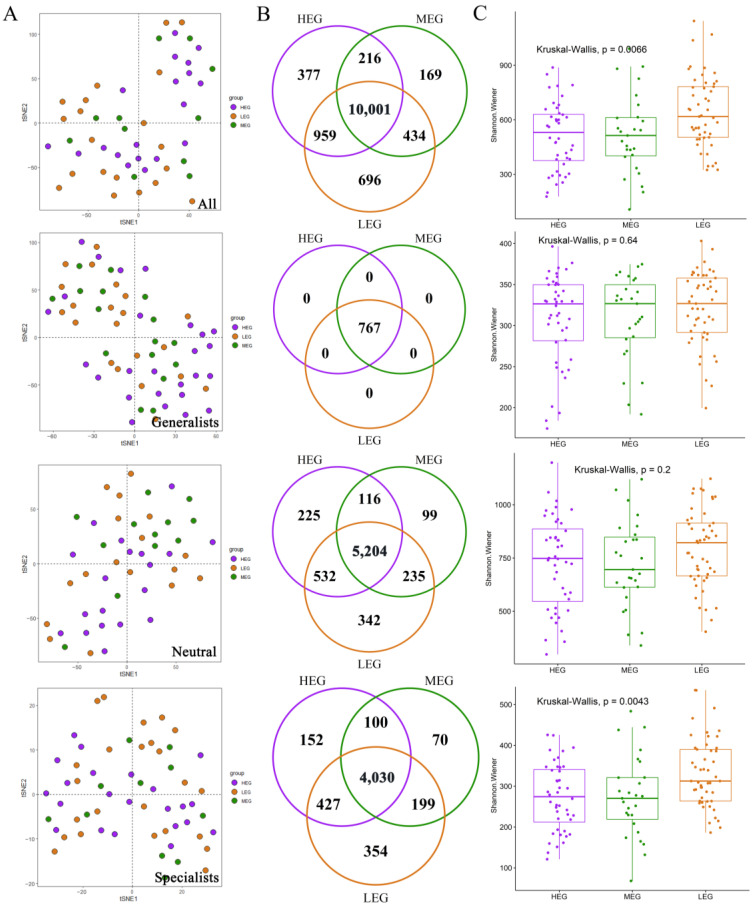
(**A**) Nonmetric multidimensional scaling (NMDS) showing the variation of soil bacterial communities across altitudes gradients. (**B**) Venn diagram showing the numbers of unique and shared OTUs among three humidity gradients. (**C**) Species richness along different altitudes. Different numbers indicate significant differences at *p* < 0.05 according to the Kruskal–Wallis test.

**Figure 4 microorganisms-12-01034-f004:**
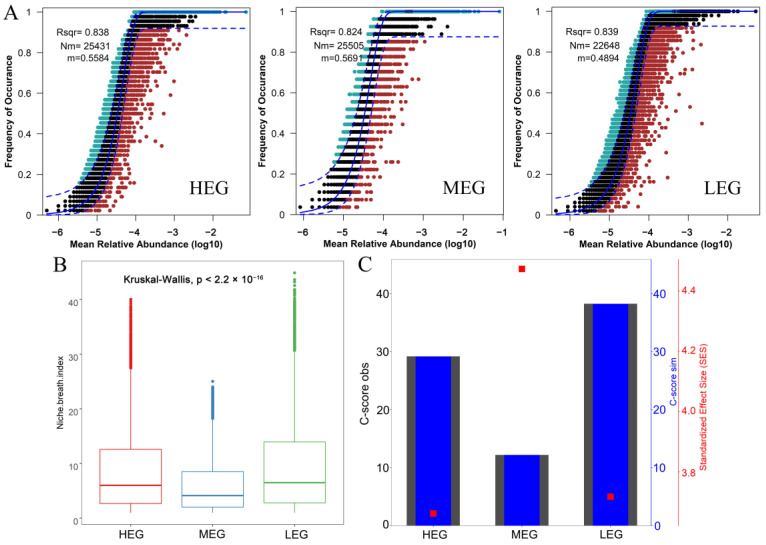
Ecological processes of the soil bacterial community. (**A**) A neutral model was used to assess the effects of random diffusion on soil bacteria. Rsqr indicates the goodness of fit to the neutral model. Nm indicates metacommunities size times immigration. m indicates the estimated migration rate. The solid blue lines indicate the best fit to the neutral model and dashed blue lines represent 95% confidence intervals around the model prediction. (**B**) Comparison of niche widths of bacterial groups under different altitude gradients (*p* < 0.05 by Kruskal–Wallis test). (**C**) Evaluating the effects of deterministic processes on bacterial communities using a null model. The values of observed C-score (C-scoreobs) > simulated C-score (C-scoresim) indicate non-random co-occurrence patterns. Standardized effect size <− 0 and > 0 represent aggregation and segregation, respectively.

**Figure 5 microorganisms-12-01034-f005:**
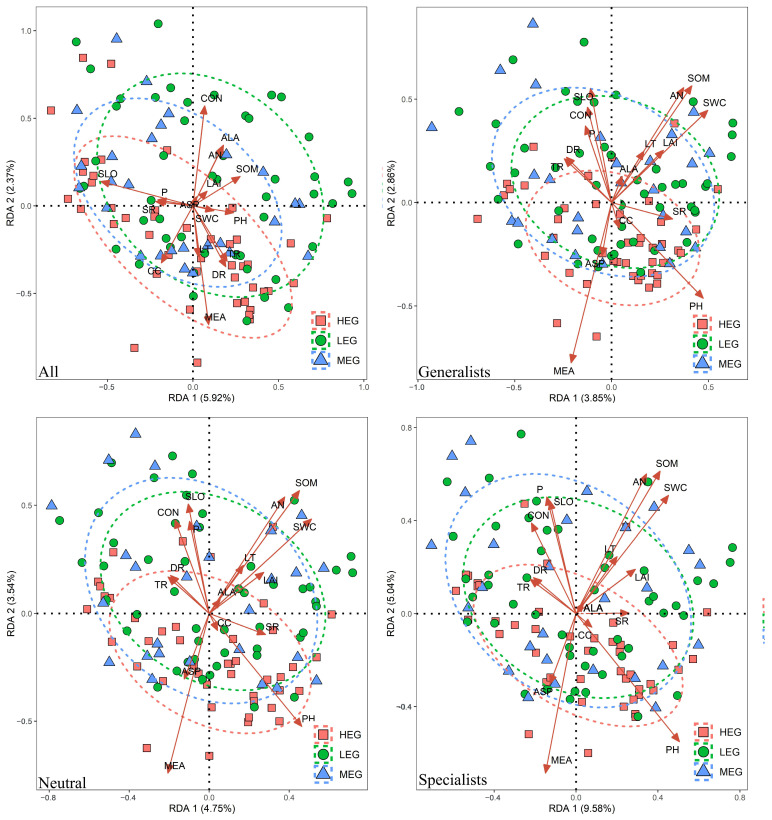
RDA analysis of soil bacterial community structure and environmental variables.

**Figure 6 microorganisms-12-01034-f006:**
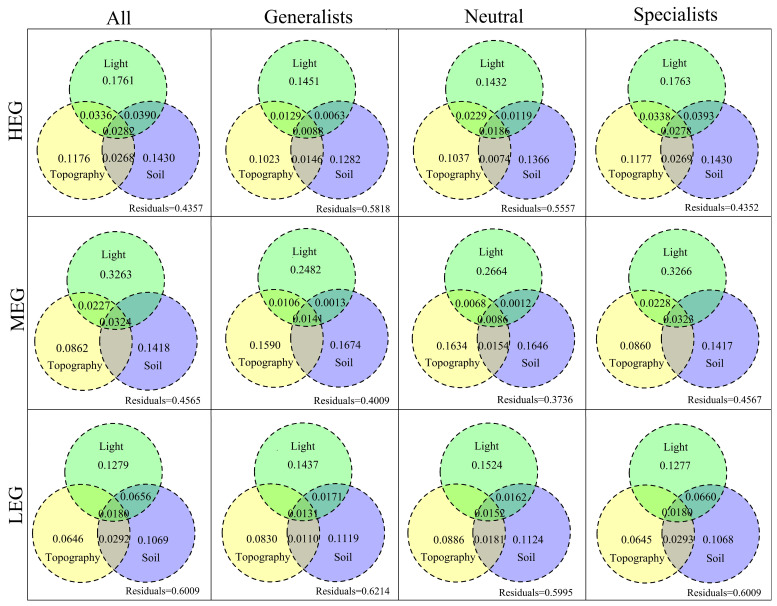
The effect of environmental factors on the variation of the soil bacterial community was revealed by variance analysis.

## Data Availability

All the raw reads of 16S genes and shotgun metagenomic sequences were deposited in the NCBI GEO under accession identification number PRJNA633088.

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
