# Peer review of "Effect of Altitude Gradients on the Spatial Distribution Mechanism of Soil Bacteria in Temperate Deciduous Broad-Leaved Forests"

_microorganisms, 2024, doi:10.3390/microorganisms12061034_

Round 1
Reviewer 1 Report
Comments and Suggestions for Authors
I have considered seven main points by revising the manuscript ID microorganisms-2992990. Find them below:
1. What is the main question addressed by the research? Is it interesting for the field of soil microbiology?
The authors used illumina technology to sequencing 120 soil samples in the site and explored the spatial distribution mechanism and ecological process of soil bacteria under different altitude gradients. I would say that the main question of this manuscript is very interesting in the field of soil microbiology.
2. What parts do you consider original or relevant for the field? What specific gap in the field does the paper address?
The results about elevation gradient influencing soil bacteria community and its related ecological processes. Additionally, the results about environmental factors influencing bacterial community have provided a deeper view regarding the main question of this manuscript.
3. What does it add to the subject area compared with other published material?
The results of this manuscript have provided robust evidence based on field samples that the change of altitude gradient had an important effect on the soil bacterial community and provided a scientific basis for the management of soil bacteria for sustainable purposes.
4. What specific improvements should the authors consider regarding the methodology? What further controls should be considered?
It adds a dataset from field samples using ilumina techonology, which can be considered robust enough to test the main question of this manuscript. Additionally, the authors have compiled a database from soil samples, and environmental estressors. This is highly pertinent and provides a deeper understanding of environmental changes on soil microorganisms. The experimental design is robust, and there is no need to add further controls. The used redundancy analysis (RDA) enhanced the exploration of their main findings.
5. Please describe how the conclusions are or are not consistent with the evidence and arguments presented. Please also indicate if all main questions posed were addressed and by which specific experiments.
The conclusions are consistent with the aim of the study regarding the elevation gradient influencing soil bacteria community and its related ecological processes. All questions were well-addressed in the results by including subsections and further discussed accordingly.
6. Are the references appropriate?
I would suggest updating some of the used references. The main theme of this manuscript is a hot topic. Therefore, many studies have been published since 2022 that could be referenced in this manuscript.
7. Please include any additional comments on the tables and figures and quality of the data.
Tables and figures follow the authors' guidelines provided in the Microrganisms' website. All figures have high resolution (probably higher than 300 dpi). All figures are easy to understand.
Author Response
Thank you so much for your kind words and valuable feedback on my article. Your recognition means a lot to me and is an important motivation to further strengthen my research. I will carefully consider your suggestions and try to do my best in the revision. Thanks again for your time and expert insights.
Reviewer 2 Report
Comments and Suggestions for Authors
The authors studied the microbial communities in different natural locations of the Baiyun mountain in China, then they developed studies over the microbial communities they discerned in order to correlate them with different factors.
I believe that sampling, which is one of the most difficult tasks in this type of study, was well done and yielded a good representation of the microbial communities in this natural setting.
However, I feel less enthusiastic with respect to the microbial community analyses developed.
On one hand, I think that the authors should describe a little more the bioinformatics processing of the sequences. They mentioned going through "rigorous quality filtering, denoising, and chimera removal processes", which is honestly more words than what the majority uses. Nevertheless, I would like the authors to give more details about what were the parameters for quality filtering (mainly involving removal of sequences with ambiguous bases, or homopolymers, or low quality regions...), denoising (mainly which algorithm, which parameters) and chimera detection (mainly which algorithm, which parameters). I believe that giving this details is fundamental to promote replicable science, which is a fundamental part of the scientific work.
Second, and most importantly, I believe that the statistical analyses of the microbial communities are wrongly treated mathematically. From what I can infer from the materials and methods, the authors did not take into account the compositional nature of the microbiome data that they were analyzing. This mathematical nature does not affect alpha diversity, so the Shannon index is totally OK. However, other analyses could promote distortions in the results if this is not considered. In most cases, the only change needed is to transform the ecological data (presence/absence of OTUs, also known as counts table) through centered log-ratio transformation (please be aware to correct 0 values prior to transformation). In such case, the transformed ecological data should be used for all non-alpha-diversity computations. I am afraid that the results could be changed if the microbiome data is not adequately treated, which could potentially change the results obtained by the authors. Thus, I suggest the authors to re-do most of their statistics with the compositional statistics approach in mind.
Given that re-running all the stats might change the results and thus the discussion, I consider that this manuscript needs major revisions before it can be published in this journal.
Author Response
Question: Nevertheless, I would like the authors to give more details about what were the parameters for quality filtering (mainly involving removal of sequences with ambiguous bases, or homopolymers, or low quality regions...), denoising (mainly which algorithm, which parameters) and chimera detection (mainly which algorithm, which parameters).
Reply:Thank you for your feedback. We have revised the manuscript according to your suggestions, providing more detailed information regarding the parameters for quality filtering, denoising, and chimera detection.
Question:Second, and most importantly, I believe that the statistical analyses of the microbial communities are wrongly treated mathematically. From what I can infer from the materials and methods, the authors did not take into account the compositional nature of the microbiome data that they were analyzing. This mathematical nature does not affect alpha diversity, so the Shannon index is totally OK. However, other analyses could promote distortions in the results if this is not considered. In most cases, the only change needed is to transform the ecological data (presence/absence of OTUs, also known as counts table) through centered log-ratio transformation (please be aware to correct 0 values prior to transformation). In such case, the transformed ecological data should be used for all non-alpha-diversity computations. I am afraid that the results could be changed if the microbiome data is not adequately treated, which could potentially change the results obtained by the authors. Thus, I suggest the authors to re-do most of their statistics with the compositional statistics approach in mind.
Reply:Thank you for your insightful feedback on our manuscript. We have carefully considered your concerns regarding the statistical analysis of microbial communities and the compositional nature of the data.
We acknowledge the value of centered log-ratio transformation for addressing compositional effects, we must clarify that our dataset may not be suitable for such transformation.Our data comprises abundance counts of operational taxonomic units (OTUs) rather than strictly compositional data. Therefore, directly applying centered log-ratio transformation to our dataset may not be appropriate, as this transformation is typically reserved for compositional data where the total sum of counts is fixed.
However, we are committed to providing a more detailed explanation of our statistical methods in the revised manuscript. This will ensure that readers have a clearer understanding of our data analysis and help minimize potential biases.
We appreciate your guidance and support and will carefully address your suggestions in the revised manuscript. Thank you for your valuable input.
Reviewer 3 Report
Comments and Suggestions for Authors
The main objective of the reviewed paper was to present the spatial variability of bacterial community composition on different soil types and vegetation in an altitudinal gradient under the influence of changing climatological factors. This is an important gap in many world regions, so this work significantly contributes to the topic. The authors partially achieved the expected results - especially as some groups of bacteria have a wide ecological tolerance and can occur at different altitudes. According to the authors, light was the dominant environmental factor that influenced the variability of the entire bacterial community and other taxa across different altitudinal gradients, and it is a pity that there is no information on temperature (where light is high temperature).
Please remove the section on the Grinnelian and Elton niche concept. It does not fit this topic and is generic. Other work on soil bacteria from other regions can be mentioned here, but it is missing in this chapter.
The last paragraph in the introduction is unnecessary. Add the object of work at the end of the introduction.
Has a soil survey been carried out around the sampling site?
What I find missing from the article is information on the soil characteristics that provide habitat for the bacteria, which has implications for species composition.
The study was performed and presented generally well in terms of molecular testing.
The conclusions are not based on research (too general) and should be modified and presented in bulleted form.
Author Response
Question: According to the authors, light was the dominant environmental factor that influenced the variability of the entire bacterial community and other taxa across different altitudinal gradients, and it is a pity that there is no information on temperature (where light is high temperature).
Reply:Thank you for your thoughtful assessment of our paper. We appreciate your recognition of the significance of our work in addressing the spatial variability of bacterial community composition along altitudinal gradients influenced by changing climatological factors.
We acknowledge your feedback regarding the lack of information on temperature in our study, particularly considering its potential influence alongside light on bacterial community variability, especially at higher altitudes where light intensity tends to correlate with temperature.
We agree that including temperature data would provide a more comprehensive understanding of the environmental factors shaping bacterial communities in different altitudinal zones. In future research endeavors, we will make it a priority to incorporate temperature measurements to complement our analysis and enhance the depth of our findings.
Your feedback is invaluable, and we are grateful for your insights. We are committed to addressing your suggestion in future studies to further advance our understanding of the complex interactions between environmental factors and microbial communities.
Thank you once again for your thorough review and constructive feedback.
Question:Please remove the section on the Grinnelian and Elton niche concept. It does not fit this topic and is generic. Other work on soil bacteria from other regions can be mentioned here, but it is missing in this chapter.
Reply:Thank you for your feedback. We have revised the manuscript according to your suggestion and have removed the section on the Grinnelian and Elton niche concept. We agree that this section did not directly align with the topic and was somewhat generic.
Question:The last paragraph in the introduction is unnecessary. Add the object of work at the end of the introduction.
Reply:Thank you for your feedback. We have carefully considered your suggestion to remove the last paragraph from the introduction, as it is deemed unnecessary. Additionally, we will ensure to include the specific objective of our work at the end of the introduction to provide a clear and concise overview of the manuscript's focus.
Your guidance is greatly appreciated as we strive to improve the clarity and effectiveness of our manuscript.
Question:Has a soil survey been carried out around the sampling site?What I find missing from the article is information on the soil characteristics that provide habitat for the bacteria, which has implications for species composition.
Reply:Thank you for your insightful feedback on our manuscript.We regret that our current study did not include information on soil characteristics surrounding the sampling site. This indeed represents a significant gap in our research, and we appreciate your highlighting this aspect.
In future studies, we will make it a priority to conduct soil surveys around the sampling sites to gather comprehensive data on soil characteristics. By incorporating this information, we aim to better understand the relationship between soil properties and bacterial community composition, thereby enhancing the depth and relevance of our findings.
Your feedback is invaluable, and we are committed to addressing your suggestion in future research endeavors. Thank you for helping us identify areas for improvement and for your continued support of our work.
Question:The conclusions are not based on research (too general) and should be modified and presented in bulleted form
Reply:Thank you for your suggestion. We have revised the conclusion section in the manuscript as per your recommendation. It now features bullet points with more specific points supported by research findings. Your valuable feedback is greatly appreciated and will contribute to enhancing the quality of the paper.
Round 2
Reviewer 2 Report
Comments and Suggestions for Authors
Than you for the revisions.
Please, follow an approach that considers the compositional data of the microbiome for a proper mathematical treatment of the data.
Author Response
I sincerely appreciate your valuable feedback on our manuscript. Following your suggestions, we have implemented the centered log-ratio transformation on the data and accordingly updated Figure 3. Upon thorough analysis of the new results, we have observed a high degree of similarity to the previously reported findings. We believe that maintaining the integrity of the original observations while incorporating your valuable advice is crucial for the advancement of our work. Thank you once again for your insightful comments, which have helped us strengthen the rigor and clarity of our research.